# Evolution-Informed Strategies for Combating Drug Resistance in Cancer

**DOI:** 10.3390/ijms24076738

**Published:** 2023-04-04

**Authors:** Kristi Lin-Rahardja, Davis T. Weaver, Jessica A. Scarborough, Jacob G. Scott

**Affiliations:** 1Systems Biology & Bioinformatics, Case Western Reserve University, Cleveland, OH 44106, USA; 2Department of Translational Hematology & Oncology, Cleveland Clinic Lerner Research Institute, Cleveland, OH 44106, USA

**Keywords:** cancer, drug resistance, evolution, ecology

## Abstract

The ever-changing nature of cancer poses the most difficult challenge oncologists face today. Cancer’s remarkable adaptability has inspired many to work toward understanding the evolutionary dynamics that underlie this disease in hopes of learning new ways to fight it. Eco-evolutionary dynamics of a tumor are not accounted for in most standard treatment regimens, but exploiting them would help us combat treatment-resistant effectively. Here, we outline several notable efforts to exploit these dynamics and circumvent drug resistance in cancer.

## 1. Introduction

Oncogenesis occurs by mutation and selection from a clonal population. The model of cancer evolution proposed by Nowell in 1976 [1] has been corroborated by more than 45 years of experimental evidence. Briefly, tumors evolve from individual cells of corresponding normal tissue by acquiring successive growth-promoting adaptations. These adaptations can be genetic or epigenetic and are caused by mutation, copy number variation, gene fusion events, and other molecular derangements. Adaptations that increase the fitness of a given cell are selected, leading to the outgrowth of clonal populations. As sequential adaptions are acquired, tumors progress from normal tissue to pre-cancerous lesions, cancer in situ, local cancer, and finally, to disseminated, metastatic cancer. At the time of diagnosis, a solid tumor is typically a highly heterogeneous population. Further, due to the differential selective pressures applied by different areas of the body, metastasis drives further differentiation. Even within a single tissue, selective pressures can vary greatly due to immune effects, normal tissue interactions, distance from blood vessels, and clonal interference [2]. These evolutionary processes enable tumors to adapt to almost any selective pressure, including clinical treatments.

The striking ability of cancer to evolve resistance against chemotherapy is one of the most daunting challenges in oncology today. Cancer is a highly diverse disease—tumors can arise from different tissue types, which can cause patients to have differing symptoms. Even people with tumors in the same organ can have variations between their symptoms and response to treatment. The severity of the cancer is measured in four stages, which can affect prognosis. Tumors can also differentiate to various degrees. The more poorly differentiated a tumor is, the more dissimilar it is to the tissue of origin, and the prognosis tends to be worse. Differences in tumors across individuals are referred to as intertumoral heterogeneity. Additionally, as cells within a tumor rapidly proliferate, subpopulations with different evolutionary trajectories can emerge, leading to intratumoral heterogeneity. Subpopulations within a tumor can have varying levels of response to treatment and can sometimes be innately resistant to certain agents, making treatment more difficult and disease recurrence more likely. Given the diversity of the disease, a variety of approaches have been taken to treat it (Figure 1). Mechanisms of drug resistance vary greatly with the nature of the drug, spatial constraints, the heterogeneity of the tumor population, and other variables within the tumor microenvironment. The appropriate method of addressing resistance, therefore, may look very different from case to case [3]. Resistance to cytotoxic chemotherapies most often arises because the drug is unable to reach the target it seeks to inhibit [4]. Some commonly characterized mechanisms include a lack of drug uptake due to altered drug metabolism in the tumor, removal of the drug from tumor cells, alteration of the drug target, or sequestering of the drug away from the target [4,5,6,7,8]. By contrast, resistance to targeted therapies, an application of precision medicine, typically emerges because components downstream of the target persist via amplification of the target, secondary mutations emerge, or downstream pathway suppressors are functionally lost [4,9,10,11,12,13]. Finally, resistance to immunotherapies, such as immune checkpoint inhibitors (ICIs) or adoptive cell therapies, occurs when tumor cells evolve to evade/suppress the immune system entirely. This is accomplished via immunoediting, where the tumor and immune system move through phases of elimination, equilibrium, and escape [14,15,16].

To combat resistance, combinations of drugs are often used so that a tumor is attacked in multiple ways at once [17,18,19]. By treating a tumor with multiple drugs, the potential to debilitate multiple molecular mechanisms within the tumor cells is increased. Most often, combinations of different cytotoxic chemotherapies are used. Targeted therapies are included if a patient tests positive for a druggable mutation. Combining immunotherapy with traditional chemotherapy is also becoming more common, especially as adjuvant therapy. Typically, ICIs form the backbone of most of these combinations [20]. Depending on the combination and the cancer type, multi-drug regimens may be administered with all drugs simultaneously, or by cycling through different drug cassettes [18,21].

As opposed to using single agents, using combinations of drugs has yielded greater benefits for cancer patients as a whole and has become standard practice for a majority of cancers. Despite this improvement, however, this strategy of attacking the tumor in many ways at once does not negate the inevitability of therapeutic resistance. While rapidly eliminating all treatment-sensitive cells initially has a positive impact on tumor burden and temporarily quells disease symptoms, treatment-resistant cells will remain alive in the population. With no sensitive cells remaining to compete for resources, the resistant cells are able to proliferate more rapidly leading to disease recurrence. This dynamic, known as competitive release, is well-documented in bacteria and other studies in ecology. It has recently been explored in cancer as well [22,23,24]. Thus, aggressive treatment strategies may actually accelerate the growth of treatment-resistant tumors [24].

Many additional strategies for tackling therapeutic resistance have been explored. Examples include targeting specific drug resistance-inducing pathways, inhibiting cancer stem cells, conjugating prodrugs with peptides to encourage activation, and using oncolytic viruses [25,26]. While these strategies are promising, they generally lack consideration for cancer’s ability to rapidly adapt to selective pressure. When clinicians overcome one resistance strategy, another mechanism is often subsequently evolved by the tumor [27]. This adaptability is the driving force behind treatment resistance.

In the concluding remarks of Nowell’s seminal paper on cancer as an evolutionary process, he said “More research should be directed toward understanding and controlling the evolutionary process in tumors before it reaches the late stage usually seen in clinical cancer.” [1]. Undoubtedly, substantial work has been performed to better understand the evolutionary processes in tumors. Some of this work has been summarized in other excellent reviews [2]. In this work, we turn our attention to the second part of Nowell’s remarks. We will highlight the progress made to date toward controlling and exploiting the evolutionary process in tumors to achieve superior clinical outcomes for patients. Finally, we will present a new framework for cancer therapy that explicitly accounts for the adaptive capacity of cancer.

## 2. Predicting Drug Sensitivity to Improve Treatment Planning

Every tumor is unique, and standard treatment strategies are not guaranteed to be effective for every individual. For instance, the majority of cancer patients will receive combinations of drugs to treat their disease. Compared to single agents, combinations have resulted in a greater overall benefit to patients [28]. This could potentially be attributed to drugs interacting with one another in a beneficial manner, namely via additive or synergistic interactions, but this has yet to be thoroughly investigated in a clinical setting [29]. The benefit of using drug combinations might also be attributed to combinations simply having a higher likelihood that any given patient will be responsive to at least one of the drugs used [30]. In the latter scenario, if a patient is resistant to one or more of the drugs being used, they would be receiving more than what is necessary for effective treatment and experience excessive toxicity with no added benefit [20,31,32]. In these cases, it would be better for patients to receive only drugs with known efficacy against their tumor so that excessive toxicity is minimized. Additionally, patients whose tumors have evolved resistance to drugs that once worked well for them may not have well-defined options for second-line chemotherapies. This circumstance is especially true for those with rarer cancers lacking extensive research, such as Ewing’s sarcoma, a rare pediatric cancer with no clear standard second-line therapy [33].

It would be worthwhile to assess which drugs are optimal for an individual prior to drug administration and after resistance has evolved to initial treatment. A clinical workflow for personalized drug regimens is already established for antibiotics. Patients needing antibiotics will often have an antibiogram performed, where a drug screen is performed on a sample or blood culture, and the caregiver provides the antibiotics shown to be impactful for that particular patient [34]. This is effective because bacteria proliferate rapidly, and antibiotics are far less expensive compared to cancer drugs. However, compared to bacteria, extensive drug screens for human tissue samples are time-consuming, error-prone, and costly, making them impractical for clinical purposes [35]. Generating accurate results could take weeks to months, and by the time a drug screen has been completed and the results sent to the clinician, the tumor may have already evolved a different response profile. Further, growing human tissues in vitro requires significant adaptation by the cancer cells, making cancer cell lines an imperfect proxy of the parent tumor. Because of these difficulties, there are many ongoing efforts to predict drug response, thereby foregoing the need for a drug screen. Predicting drug response would be less costly while still allowing caregivers to maximize treatment efficacy and minimize excessive toxicity on an individual basis. A patient’s sensitivity profile could be assessed multiple times throughout treatment. Ideally, predictions would be made prior to treatment, and this could be repeated whenever resistance emerges. By adopting this workflow, patients would be less likely to receive drugs that lack efficacy against their tumor at any time throughout treatment. Such a workflow could be referred to as a “chemogram”—just as antibiograms inform clinicians of which antibiotics to prescribe a patient, a chemogram would be a method of informing clinicians which chemotherapies are optimal for a cancer patient. Below, we outline a few promising methods for drug response prediction.

### 2.1. Gene Signatures

Gene signatures are defined as a set of genes whose expression is associated with a particular phenotype. Prognostic gene signatures are used to assess the severity of a patient’s disease, while predictive gene signatures are used to predict the response to a particular therapy. Prognostic gene signatures have transformed clinical practice in breast cancer. Examples include Mammaprint and Oncotype DX, which project the risk level for breast cancer recurrence [36,37]. Similarly, the decipher genomic classifier is a biomarker-based tool for the prognosis of prostate cancer [38]. Predictive gene signatures have yet to be used in a clinical setting, but there are several pre-clinical signatures demonstrating potential benefits for predicting therapeutic responses in cancer patients. For instance, the radiation sensitivity index (RSI) [39,40] has been shown to be a useful method for delineating patients who are relatively resistant or sensitive to radiation therapy in a number of disease sites, such as lung, melanoma, glioma, head and neck, and others [41,42,43,44,45,46,47,48,49,50,51,52]. In an extension of the RSI, a genomically adjusted radiation dose (GARD) predicts the therapeutic effect of a given radiation dose by combining the aforementioned signature with a linear-quadratic model [46,53]. Gene signatures are also promising tools for integrating evolutionary concepts with the prediction of a therapeutic response. For example, Scarborough et al. designed a method of extracting signatures inspired by the concept of phenotypic convergent evolution, where similarities in gene expression among genomically disparate cancer cell lines exhibiting sensitivity to cisplatin were identified and used as a predictive signature [54]. Similarly, a study on Ewing’s sarcoma sought to capture convergent states of collateral sensitivity in treatment-resistant tumors, and from this, generated biomarkers based on the similar expression patterns identified across many biological replicates [55].

With further research, predictive gene signatures may become an invaluable tool in oncology and could be an integral device for the development of chemograms in cancer. Eventually, they could be used to help clinicians optimize treatment plans on an individual basis by assessing drug concentrations during therapy [56] and determining if therapeutic resistance has evolved [57]. Again, it is critical that generating a patient’s sensitivity profile is performed quickly so that their tumor does not evolve a new profile by the time treatment is administered. Using gene expression signatures to predict a response would take far less time and financial expense than a drug screen and can also be performed at scale.

### 2.2. Computational Models

As bioinformatics and computational biology have continued to expand in their utility, a growing number of computational models have been designed to predict a drug response. Currently, all advances in this area of research are confined to the pre-clinical stage. Models range from relatively basic, such as the gene signatures described previously [39,46,54,55], to highly complex, such as neural networks that integrate many different molecular features [58,59,60]. Gene expression is typically the major basis of response prediction.

The complexity of computational models has increased dramatically in the past few decades, and with it, the potential for more accurate outcome predictions. To compare prediction accuracy among three common machine learning models, one study used a random forest model, an elastic net, and a deep neural network. The models were trained on data from the Genomics of Drug Sensitivity in Cancer (GDSC), a commonly utilized public repository containing gene expression and drug response data for 1001 cell lines across different cancer types [58]. Across multiple disparate clinical datasets, including The Cancer Genome Atlas (TCGA), the deep neural network consistently predicted drug response with higher accuracy than the other two models. Chawla et al. further demonstrated the utility of deep neural networks for drug response prediction by using gene expression, pathway enrichment and drug features to train their model, named Precily [59]. While Precily was unable to predict the exact drug response values in a prostate cancer cell line treated with two different drugs, it was able to capture the relative difference in sensitivity. Using Precily to categorize TCGA patients into two groups based on predicted survival showed improved survival in the patients predicted to be drug sensitive. In the interest of capturing tumor heterogeneity in a more explicit manner, another study utilized single-cell RNA-seq data and matrix factorization to predict drug response [60]. The recommender system was trained on the GDSC dataset with cross-validation and tested with numerous patient-derived cell lines. The system separated sensitive cells from resistant ones and predicted the sensitive cohorts’ IC50s with 80% accuracy. Finally, the model was able to predict complementary combinations of drugs and how much benefit they may confer over monotherapy. Predictions were validated by testing five different drug combinations against five patient-derived cell lines. The model did not account for the possibility of drug interactions but still yielded a notable positive correlation between predicted and observed responses. This capability is especially noteworthy, given that most patients receive multiple drugs at once. There are many more examples of computational prediction models, all using various models and unique approaches [61,62,63,64,65]. With any given method of drug response prediction, it will be key to adjust the regimen as treatment proceeds, based on what a tumor’s molecular profile reflects as it continues to evolve and change.

The potential for using computational models to predict drug response is undoubtedly promising, but progress is limited by the availability of appropriate data types to train these models on. For instance, GDSC is widely used for this purpose but only contains data from cell lines, which greatly hinders clinical translation [66]. Additionally, clinical trial data containing outcomes and molecular characterization of tumors are also available, but potential applications outside of the initial study tend to be more limited because trials typically focus on one specific treatment regimen. Another example is TCGA, an incredibly useful public repository containing a large variety of data types collected from thousands of patients with different types of cancer [67]. A key limitation of this dataset is the lack of robust reporting on which treatments individual patients received. What these repositories lack, as well as most other public data sources, is a continuous temporal and spatial aspect that reflects the constant evolution of a tumor throughout time. Most data repositories typically only report data for the beginning and end of an experiment, but what happens in between is rarely documented. Characterizing the molecular profile of a tumor as it changes over time would be highly informative, especially if the tumor is evolving therapeutic resistance, and experimental evolution on cancer cells would generate such data. This information could potentially be used to accurately forecast when resistance will occur before it begins to affect a patient’s disease symptoms. In doing so, we could minimize or potentially eliminate periods of time where a patient is receiving ineffective treatment and experiencing excessive toxicity.

### 2.3. Experimental Evolution and Collateral Sensitivity

Performing experimental evolution in cancer presents an opportunity for researchers to more thoroughly consider how the continuous evolution of a tumor impacts treatment response. Experimental evolution has been used to explore evolutionary dynamics in bacteria, cancer, and multicellular organisms. An outstanding demonstration of this approach is a project led by Richard Lenski, who has maintained 12 independent populations of *Escherichia coli* since 1988 and continues to this day [68]. These replicates originated from a single strain and have been evolving as entirely separate populations. This experiment, referred to as the long-term evolution experiment (LTEE), has inspired similar research in a variety of fields, including oncology. One such experiment was performed by culturing TP53-deficient human gastric organoids for two years in order to investigate stages of tumorigenesis [69]. Experimental evolution can also be employed to explore drug resistance and help inform potential treatment strategies when combined with collateral sensitivity [55,70,71,72]. This is a concept where cells, while evolving resistance to one agent, simultaneously develop sensitivity to other agents that are not involved in the active treatment. Conversely, collateral resistance (or cross-resistance) occurs when evolving resistance to one agent confers resistance to other drugs that are not included in the active treatment. A study on Ewing’s sarcoma, a rare pediatric cancer that lacks standard second-line therapy, combined these ideas by evolving therapeutic resistance against the first-line therapy for this disease [55]. Throughout the experiment, gene expression was measured, and the samples were screened against nine different drugs to determine if there were any collateral responses. The experiment involved seven biological replicates of a Ewing’s sarcoma cell line, and parallels in collateral response across these samples were identified so that predictive biomarkers could be extracted from the gene expression data. The results of this evolution experiment showed collateral sensitivity repeatedly evolving against one of the nine drugs, suggesting that it may be worthwhile to further investigate its potential as second-line therapy for this disease. Collateral resistance was also observed for three of the drugs across most of the replicates, indicating that these drugs may not be useful as second-line therapies for treatment-resistant Ewing’s sarcoma. This is but one example of the multitude of possibilities that experimental evolution in cancer presents.

Considering that cancer is actively evolving from the moment of tumorigenesis, the use of experimental evolution would be a relevant approach for understanding any stage of the disease. To further enhance our understanding of the evolution of drug resistance in cancer, technologies, such as CRISPR/Cas, clonal tracing, or spatial analyses, can be applied to experimental evolution. The CRISPR/Cas system can be used to characterize potentially different mechanisms of resistance against a drug [73]. For instance, this could be applied to the Ewing’s sarcoma experiment described above. As resistance to standard chemotherapy evolves in independent biological replicates, these disparate populations may be using different mechanisms to enable resistance. A CRISPR/Cas knockout screen could be used to identify differences or similarities among the resistance mechanisms used by the independently evolving populations, thereby revealing potential targets for treatment after resistance has evolved.

Another useful tool for furthering our understanding of resistance evolution is clonal tracing. This experimental method can be used to examine the evolutionary dynamics of evolving cells by inserting heritable barcode tags in individual cells within a starting population and characterizing phenotypic changes in the subpopulations that arise from different clones [74]. Such phenotypes could include changes in the transcriptome, proteome, and drug responses. Spatial omics analyses would further enhance the resolution at which we can describe subpopulations that arise from these clones and the evolutionary dynamics that occur within and among them over time. Clonal tracing has been used to identify the origin and evolutionary process of resistance to targeted therapies in ALK+ non-small cell lung cancer [75]. This study, among others, found evidence that resistance can emerge de novo from reversibly drug-tolerant persister cells [75,76,77]. These cells are found to be weakly resistant to treatment, dynamically regulated depending on the presence or absence of a drug, and a substrate for the evolution of therapeutic resistance to progress as treatment continues. Thorough investigations of the origins and progression of drug resistance would reveal possible avenues for evolution-informed therapy, which anticipates the emergence of resistance, rather than responding to it after it has already arisen.

## 3. Controlling Evolution by Exploiting Ecological Dynamics

Tumor heterogeneity is widely recognized as a key driver of treatment resistance [78,79]. The presence of many subpopulations in a tumor makes it difficult to determine if a particular drug will have a substantial impact on the tumor as a whole. The greater a tumor’s heterogeneity is, the more opportunities there are for acquired resistance to emerge or for innately resistant populations to exist already. There is a growing interest in understanding how these subpopulations interact with one another [80,81,82,83,84]. The field of cancer ecology employs eco-evolutionary principles to understand the dynamics of carcinogenesis, tumor progression, and response to treatment. This has led researchers to apply ecological models to cancer in order to control the rate and trajectory of evolution and avoid therapeutic resistance altogether [85,86,87,88,89,90].

A common concept in ecology is the principle of fitness trade-offs, where organisms evolving selective advantages to one set of selective pressure will also evolve disadvantages to different sets of selective pressures. Collateral sensitivity is an example of a fitness trade-off: Evolving resistance to treatment is a benefit to fitness, but the cost of this gain is susceptibility to a different form of treatment [91]. Fitness costs can also be observed as a decrease in growth rate. For instance, monoculture of gefitinib-resistant non-small cell lung cancer (NSCLC) cells in the absence of the drug have a lower growth rate than gefitinib-sensitive cells [23].

Competitive exclusion is another ecological concept that can be applied alongside the concept of fitness trade-offs and has only recently been elucidated in cancer [23,92,93,94,95]. This principle states that two populations competing for the same resources cannot coexist at constant population values, and one population will have an advantage over the other. Competitive exclusion is demonstrated in the same study described above, where coculturing of gefitinib-resistant and gefitinib-sensitive NSCLC cells in the absence of gefitinib resulted in the competitive exclusion of the resistant cells. Conversely, introducing gefitinib to the coculture caused the sensitive cells to be competitively excluded instead, allowing for competitive release to occur in the resistant population.

The competition between sensitive and resistant populations is not taken into account for most chemotherapy regimens, where the goal is to kill as many cells as possible by administering the maximum tolerated dose (MTD) [24,96,97,98]. In light of the concepts outlined above, this strategy would actually promote the rapid proliferation of resistant cells and subsequently, treatment failure.

### Adaptive Therapy

Adaptive therapy is an approach to treatment based on maintaining a proportion of a treatment-sensitive population and including treatment holidays, where no drug is administered [22]. While the tumor will not be eliminated, its mass can be reduced and stabilized so that disease symptoms can be relieved to some extent, and the competitive release of the resistant population can be circumvented entirely (Figure 2) [23]. Different variables can be adjusted to achieve this, such as the drug dosage, duration and frequency of treatment, and the drugs administered. Adaptive therapy with changing dosage, duration, and frequency has been tested via computational models, monoculture and coculture in vitro experiments, and in clinical trials [99,100,101]. These studies used on-off dosing regimens, and treatment holidays were implemented according to features of the tumor (such as transcriptional states and tumor volume).

In a study using the MCF7 human breast cancer cell line, doxorubicin-resistant and doxorubicin-sensitive cells were cocultured and treated with either a continuous or adaptive therapy regimen [22,88,99,100,101,102,103,104]. With the adaptive treatment, a treatment holiday began after the population fell below 50% and continued until the population regained its original size. This experiment continued until the sample was cured (no remaining cells), recurred (population size reached 4/3 of the initial size), or reached two years of treatment. The two therapy regimens were tested against various proportions of sensitive to resistant cells, and results within each treatment varied with these proportions. Samples with primarily sensitive cells were curable with continuous treatment, but samples with high heterogeneity recurred within a year using the same treatment. With adaptive therapy, samples that were heterogeneous with a larger proportion of sensitive cells eventually recurred because the sensitive population evolved resistance, but recurrence was delayed compared to continuous treatment. In samples with high heterogeneity, adaptive therapy delayed recurrence until the experiment reached two years of age. Another study, focused on BRAF-mutant melanoma, used mathematical models to determine treatment scheduling for mice with patient-derived xenografts, and adaptive treatment was adjusted based on tumor volume every 2–3 days over the course of 37 days. The adaptive schedule significantly improved the stabilization of tumor volume compared to both continuous treatment and a fixed treatment holiday schedule [100]. In a pilot clinical trial of 11 patients with metastatic castrate-resistant prostate cancer, the initial tumor burden was measured prior to treatment, and abiraterone was administered until the tumor burden was reduced to 50% of the initial baseline. Then, a treatment holiday was initiated and lasted until the baseline was restored. With this adaptive regimen, 10 of the 11 individuals maintained stable tumor burdens and the median time to progression was 27 months [101]. Compared to the median time to progression of 16.5 months with standard dosing, the results from the adaptive therapy pilot trial were promising.

There were several similarities across all of these experiments. Notably, continuous treatment was highly effective for homogenous samples, but not for heterogeneous ones. Additionally, the adaptive treatment schedules always varied from individual to individual. This further supports the claim that because intertumor composition is variable a one-size-fits-all approach is not an optimal treatment strategy. Highly homogenous tumors are more likely to respond well to a continuous MTD strategy, while those with heterogeneous tumors would benefit more from adaptive therapy, so resistant populations are competitively suppressed. Ultimately, it is clear that considering these dynamics of competition and adjusting treatment on an individual basis has immense potential to improve survival outcomes for patients. Adaptive therapies may be further improved by exploiting collateral sensitivities as they evolve. Alternatively, rather than using treatment holidays to maintain a stable tumor population, we could use prediction models to foretell a series of drugs a tumor will evolve collateral sensitivity to, then use these drugs in succession to eliminate the tumor. This idea has not been thoroughly explored in the context of cancer yet, but it has been demonstrated in theoretical models and through experimental evolution with bacteria [105,106,107,108].

## 4. Clinical Prospect of Evolution-Informed Treatment Strategies

Drug response prediction and adaptive therapy could be integrated into standard clinical practice in a few ways. As an example, we propose the pipeline illustrated in Figure 3. First, a patient’s tumor biopsy would be assessed for the presence of any druggable mutations in order to determine if they are eligible for targeted therapy. This is typically performed using next-generation DNA sequencing (NGS) to assess the entire genome at once or using polymerase chain reaction (PCR) to screen for specific genes. Fluorescence in situ hybridization (FISH) or immunohistochemistry (IHC) can also be used for this purpose. Next, gene expression would be measured to predict which other drugs the patient’s tumor is likely to be sensitive to. Bulk RNA-seq would be suitable for this purpose, as most predictive models are trained on this type of data. If the individual harbors targetable mutations, the corresponding drugs that target these mutations can be included in the treatment. Finally, tumor heterogeneity would be quantified to determine whether the patient would benefit more from an MTD approach or from adaptive therapy. Intratumoral heterogeneity can be assessed in various ways. For instance, biopsies can be taken from multiple regions of a tumor and analyzed for variations in morphology (via radiomics or histology), genetic diversity (via DNA sequencing), or transcriptomic diversity (via RNA sequencing). Single-cell RNA-seq may be informative as well but is a rather expensive analysis to perform, which may affect clinical feasibility. It may be necessary to determine an appropriate threshold for what degree of heterogeneity is required for a patient to receive adaptive therapy over MTD, and this could vary from cancer type to cancer type. From these determinations, we could elucidate what the ideal treatment strategy is for any given patient. Furthermore, if treatment efficacy begins to decline due to therapeutic resistance, these factors could be reassessed, and the treatment could be adjusted accordingly. As with any new therapy or treatment regimen, rigorous testing is necessary before clinical testing is initiated.

## 5. Conclusions

One of the most difficult challenges in the treatment of cancer is the evolution of therapeutic resistance. Clinical oncologists and researchers alike have faced this obstacle since the advent of chemotherapy. Despite the understanding that resistance occurs due to tumor evolution, eco-evolutionary dynamics of tumors are not accounted for in most standard treatment regimens. However, considering them may help us approach treatment-resistant tumors more effectively. Computational models and predictive biomarkers, such as gene signatures, can help us form treatment plans by predicting what a tumor would be responsive to, and these models can be made more robust by exploiting convergent evolution. Additionally, being able to identify states of collateral sensitivity would help us form treatment plans specifically for those who have already evolved resistance. Adaptive therapy is formulated around the concept of maintaining competitive suppression of resistant cells by maintaining a sensitive population to compete with it. Preliminary studies in silico, in vitro, and in vivo have shown in multiple heterogeneous cancers that tumor burden is much more controlled with adaptive therapy compared to continuous treatment or fixed treatment holidays. The results from these various studies have been very encouraging, and more work is necessary to bring these ideas to a clinical setting. As the field of computational biology continues to thrive and experimental methods are more thoroughly explored and optimized from bench to bedside, we believe that applying ecological and evolutionary principles to cancer will greatly benefit all cancer patients.

## Figures and Tables

**Figure 1 ijms-24-06738-f001:**
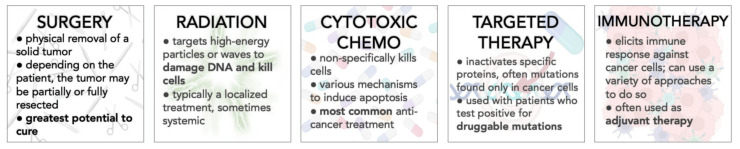
**Overview of current common approaches to cancer treatment.** Treatment strategies are chosen for a patient depending on the location and stage of the tumor. Additionally, tumors are often tested for targetable mutations to determine whether the patient is eligible for targeted therapy or not. Immunotherapy is often used as adjuvant therapy.

**Figure 2 ijms-24-06738-f002:**
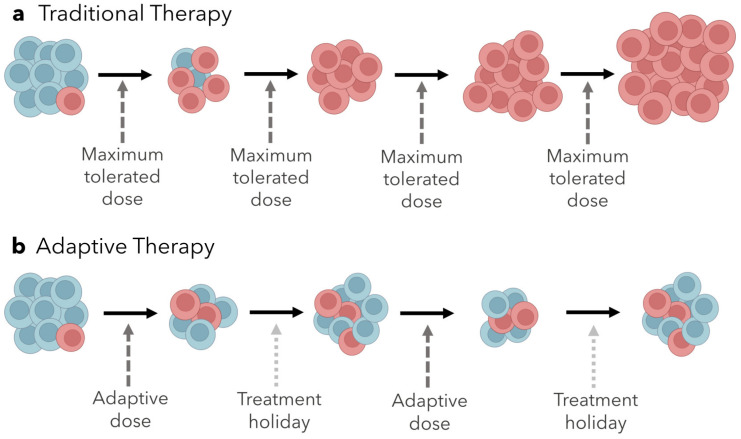
**Tumor volume over time with traditional MTD approach compared to adaptive therapy approach.** (**a**) Treating a tumor with the maximum tolerated dose initially relieves tumor burden until treatment-resistant cells evolve, eventually resulting in the competitive release of resistant cells and treatment failure. (**b**) Including treatment holidays to allow treatment-sensitive tumor cells to proliferate maintains competitive suppression of treatment-resistant tumor cells, resulting in a controllable tumor volume.

**Figure 3 ijms-24-06738-f003:**
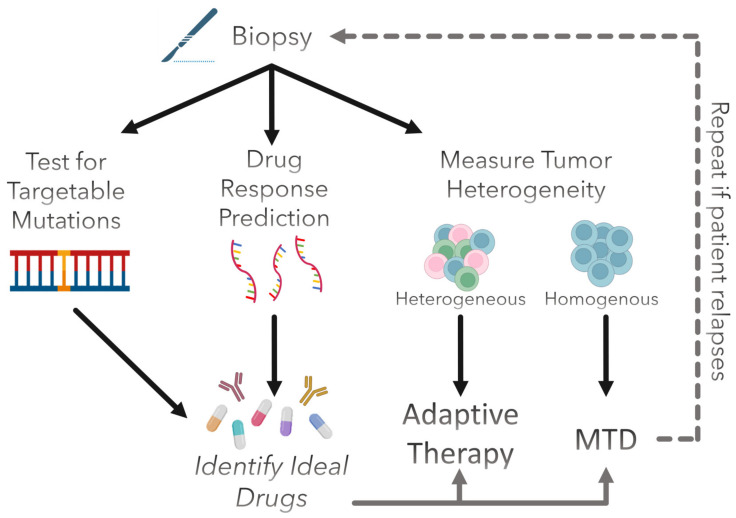
**Proposed integration of evolution-informed treatment with standard clinical practice.** A patient’s biopsy is tested for targetable mutations to determine if they are eligible for targeted therapy. Depending on this, certain drugs may or may not be included in their treatment. To further identify the optimal drugs for the patient, drug response is predicted using gene expression, perhaps through predictive gene signatures or a computational model. Based on these results, the patient could receive cytotoxic chemotherapy, targeted therapy, immunotherapy, or a combination of different kinds of drugs. Lastly, tumor heterogeneity is assessed to determine if the patient should receive adaptive therapy or traditional MTD therapy. With the optimal regimen identified, the clinician can administer the drugs predicted to have the greatest efficacy by the predictive model. The process can be repeated if the patient relapses, and a new optimal treatment could be identified.

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
