# Peer review of "Evolution-Informed Strategies for Combating Drug Resistance in Cancer"

_ijms, 2023, doi:10.3390/ijms24076738_

Round 1

Reviewer 1 Report

Lin-Rahardja et al., reviewed the eco-evolution and drug resistance in cancer. This is an interesting topic. This review manuscript would be improved if the authors could review a little bit more deep and add a perspective section showing how CRISPR, spatial technologies, clonal tracing, cancer-associated fibroblasts, an so on, can advance our understanding of drug resistance and develop new therapeutics.

Author Response

We thank you for your feedback on our review paper. We have considered your insights and decided to make additions to our review as per your suggestions. We have included perspective on how current technologies, such as CRISPR, clonal tracing, and spatial omics, can be utilized alongside experimental evolution to advance our understanding of the origins and progression of drug resistance. We also discuss the ways in which such studies may reveal potential therapeutic targets after resistance has evolved, or lead researchers to develop evolution-informed therapies that anticipate resistance and circumvent it. These revisions can be found at the end of section 2.3 "Experimental Evolution & Collateral Sensitivity".

Reviewer 2 Report

Review of the manuscript ‘Evolution-informed strategies for combating drug resistance in cancer’ submitted to the International Journal of Molecular Science by Kristi Lin-Rahardja, Davis Weaver, Jessica Scarborough, and Jacob G Scott

No new ideas are presented in this study, however interesting old ones are stressed, as are the interest of ICI immunotherapy (about which nothing is said of its frequent, sometimes letal, myotoxicities), the expected failure of treatments by chemotherapies at MTD, adaptive therapy, and the interest of drug holiday in preventing the emergence of induced drug resistance in cancer cell populations.

No explanations nor even speculations are proposed to take into account observations on evolution of cancer cell populations in Petri dishes or with organoids. Besides, the word ‘evolution’ is misleading, being used here as if evolution of a cancer cell population in a Petri dish were the only meaning of this word, while a lot of observations at the level of genes (Trigos et al. 2017,2018,2019) or in phylostratigraphic analyses (Domazet-Loso and Tautz 2008, 2010) have shown the interest of thinking cancer in the light of the great evolution of species towards and within multicellularity. I am here obviously alluding at the atavistic theory of cancer (Davies and Lineweaver Phys. Biol. 2011, Vincent Bioessays 2011), which is completely disregarded by the authors. No wonder, they start by citing Peter Nowell, and apparently consider that the acquisition of successive growth-promoting adaptations is just normal, requiring no explanation. Similarly, poor differentiation control, a characteristic of tumors (Marta Bertolaso, Philosophy of cancer 2016), is mentioned without mentioning de-differentiation nor transdifferentiation, even less plasticity. Fundamental studies on the reversibility of drug resistance (e.g., Sharma et al. Cell 2010 on persister cells), that might give food for thought to explain the interest of drug holiday (and re-challenge) in cancer treatments, are not mentioned.

This study is very shallow, possibly reflecting the limited level of thought, and daring in thinking, by the authors on cancer as a challenging disease, that requires more in-deep thinking, especially when mentioning ‘evolution’ in its misleading title, to propose new tracks for treatments.

This being said, assuming that its readers will be already convinced and satisfied with this sort of common knowledge on cancer, others like myself disregarding this study as bringing nothing new to the field, the few good things presented - although without any reasoned connecting thread - in it, progress of immunotherapies by ICIs, the expected eventual failure of drug therapies at MTD, and clinical trials and lab experiments presently conducted in the framework of  adaptive therapy (e.g., at Moffitt), deserve to be mentioned in a publication, and I will not oppose the publication of this manuscript.

In summary: acceptable for publication in its present form.

Author Response

Point 1: No new ideas are presented in this study, however interesting old ones are stressed, as are the interest of ICI immunotherapy (about which nothing is said of its frequent, sometimes letal, myotoxicities), the expected failure of treatments by chemotherapies at MTD, adaptive therapy, and the interest of drug holiday in preventing the emergence of induced drug resistance in cancer cell populations.

Response: In this review paper, we do present a few new concepts. One being the idea of a chemogram, which we defined as a framework for predicting chemotherapeutic response to a variety of drugs for individual patients, similar to antibiograms performed for patients needing antibiotics (see line 127-129). The other concept we introduce is the proposed clinical workflow illustrated in Figure 3 and described in section 4 "Clinical Prospect of Evolution-Informed Treatment Strategies".

Point 2: No explanations nor even speculations are proposed to take into account observations on evolution of cancer cell populations in Petri dishes or with organoids. Besides, the word ‘evolution’ is misleading, being used here as if evolution of a cancer cell population in a Petri dish were the only meaning of this word, while a lot of observations at the level of genes (Trigos et al. 2017,2018,2019) or in phylostratigraphic analyses (Domazet-Loso and Tautz 2008, 2010) have shown the interest of thinking cancer in the light of the great evolution of species towards and within multicellularity. I am here obviously alluding at the atavistic theory of cancer (Davies and Lineweaver Phys. Biol. 2011, Vincent Bioessays 2011), which is completely disregarded by the authors. No wonder, they start by citing Peter Nowell, and apparently consider that the acquisition of successive growth-promoting adaptations is just normal, requiring no explanation.

Response: These insights are very thought-provoking and certainly important considerations! We thank the reviewer for these comments. However, we feel it is beyond the scope of our review. We chose to focus specifically on treatment strategies that utilize cancer evolution to combat drug resistance, rather than diving into the evolutionary dynamics of the disease.

Point 3: Similarly, poor differentiation control, a characteristic of tumors (Marta Bertolaso, Philosophy of cancer 2016), is mentioned without mentioning de-differentiation nor transdifferentiation, even less plasticity. Fundamental studies on the reversibility of drug resistance (e.g., Sharma et al. Cell 2010 on persister cells), that might give food for thought to explain the interest of drug holiday (and re-challenge) in cancer treatments, are not mentioned.

Response: Thank you for this suggestion. The phenotypic plasticity of cancer cells and the studies on reversibly drug tolerant persister cells are certainly worth discussing in the review, and we have made additions to address these topics in our revisions. These additions can be found at the end of section 2.3 "Experimental Evolution & Collateral Sensitivity".